# Tadr is an axonal histidine transporter required for visual neurotransmission in *Drosophila*

Yongchao Han[1,2], Lei Peng[1,3], Tao Wang[1,2]*

[1]National Institute of Biological Sciences, Beijing, China; [2]Tsinghua Institute of Multidisciplinary Biomedical Research, Tsinghua University, Beijing, China; [3]College of Biological Sciences, China Agricultural University, Beijing, China

**Abstract** Neurotransmitters are generated by de novo synthesis and are essential for sustained, high-frequency synaptic transmission. Histamine, a monoamine neurotransmitter, is synthesized through decarboxylation of histidine by histidine decarboxylase (Hdc). However, little is known about how histidine is presented to Hdc as a precursor. Here, we identified a specific histidine transporter, TADR (torn and diminished rhabdomeres), which is required for visual transmission in *Drosophila*. Both TADR and Hdc localized to neuronal terminals, and mutations in *tadr* reduced levels of histamine, thus disrupting visual synaptic transmission and phototaxis behavior. These results demonstrate that a specific amino acid transporter provides precursors for monoamine neurotransmitters, providing the first genetic evidence that a histidine amino acid transporter plays a critical role in synaptic transmission. These results suggest that TADR-dependent local de novo synthesis of histamine is required for synaptic transmission.

## Editor's evaluation

Han et al., report the discovery of an amino acid transporter that is required locally at axon terminals of fly photoreceptors neurons for the uptake of histidine, the precursor of the neurotransmitter histamine. This function is required for transmitter synthesis locally and neurotransmission. The work exemplifies a specialized model for local monoamine transmitter synthesis at synapses in the nervous system that may set the stage for tests of generality for other monoamine systems.

*For correspondence:
wangtao1006@nibs.ac.cn

Competing interest: The authors declare that no competing interests exist.

## Introduction

Monoamine neurotransmitters including dopamine, serotonin, and histamine are formed primarily by the decarboxylation of amino acids (*McKinney et al., 2001*; *Watanabe et al., 1984*). Deficiencies in the biosynthesis of monoamine neurotransmitter such as dopamine contribute to a range of neurological disorders, such as dystonic and parkinsonian syndromes (*Kurian et al., 2011*). It has been proposed that precursor amino acids are taken up into synaptic terminals by specific transporters, followed by the synthesis and packaging of neurotransmitters within the nerve endings (*Bellipanni et al., 2002*; *Hansson et al., 1999*; *Lebrand et al., 1996*). However, to date, amino acid transporters specific for the synthesis of monoamine neurotransmitters have not been identified. Moreover, biosynthetic enzymes involved in the synthesis of neurotransmitters localize to both the soma and axonal terminals. Thus, it is possible that neurotransmitters are made in the cell body of presynaptic cells and then packed into synaptic vesicles and transported to axonal terminals via fast axonal transport (*Broix et al., 2021*; *Roy, 2020*).

Histamine was first identified as a neurotransmitter that localized to the tuberomamillary nucleus where it was synthesized from the amino acid histidine through a reaction catalyzed by the enzyme histidine decarboxylase (Hdc), which removes a carboxyl group from histidine (*Taguchi et al., 1984*; *Watanabe et al., 1984*). As a neurotransmitter, histamine plays important roles in regulating multiple physiological processes, including cognition, sleep, synaptic plasticity, and feeding behaviors (*Bekkers, 1993*; *Haas et al., 2008*; *Huang et al., 2001*; *Parmentier et al., 2002*; *Vorobjev et al., 1993*). Disruption of histaminergic neurotransmission is associated with a range of neurological disorders, including schizophrenia and multiple neurodegenerative diseases (*Haas and Panula, 2003*; *Klaips et al., 2018*; *Lim and Yue, 2015*; *Olivero et al., 2018*; *Panula and Nuutinen, 2013*; *Wang et al., 2017*). Moreover, loss-of-function mutations in the HDC gene lead to Tourette syndrome, a neurological disorder characterized by sudden, repetitive, rapid, and unwanted movements in both human patients and mouse models (*Baldan et al., 2014*; *Ercan-Sencicek et al., 2010*). However, a specific histidine transporter that maintains the histidine pool and delivers histidine to synaptic Hdc for histamine synthesis has not been identified.

*Drosophila* photoreceptor cells use histamine as the dominant neurotransmitter to convey visual signals. Thus, generation of high levels of histamine in photoreceptor neuronal terminals is important for rapid and high-frequency visual signaling (*Borycz et al., 2002*; *Hardie, 1989*; *Stuart, 1999*; *Wang and Montell, 2007*). Similar mechanisms of histamine synthesis, storage, and release between mammals and flies make the fly a powerful molecular-genetic system for studying the metabolism of neuronal histamine (*Burg et al., 1993*; *Chaturvedi et al., 2014*; *Deshpande et al., 2020*; *Gengs et al., 2002*; *Gisselmann et al., 2002*; *Hardie, 1989*; *Martin and Krantz, 2014*; *Wyant et al., 2017*; *Xu and Wang, 2019*). Histamine signals are enriched in photoreceptor terminals, and disrupting histamine synthesis by *Hdc* mutation results in reduced levels of axonal histamine and loss of visual transduction. This indicates that histamine is synthesized directly within photoreceptor terminals (*Chaturvedi et al., 2014*; *Melzig et al., 1996*). In support of this notion, LOVIT, a new vesicular transporter required for the concentration of histamine in photoreceptor terminals, is exclusively found in synaptic vesicles at photoreceptor terminals (*Xu and Wang, 2019*).

Given the high demand for histamine to maintain visual transmission at high frequencies, we hypothesized that a histidine-specific transporter must localize to neuronal terminals, and that this transporter would be required for de novo synthesis of histamine through Hdc. In support of this hypothesis, we found that Hdc localized exclusively to photoreceptor terminals. We performed a targeted RNAi screen for transporters involved in visual transmission and identified TADR (torn and diminished rhabdomeres), a plasma membrane transporter capable of transporting histidine into cells. TADR localized predominantly to photoreceptor terminals and specifically transported the amino acid histidine. Mutations in the *tadr* gene disrupted photoreceptor synaptic transmission, phototaxis behaviors, and levels of axonal histamine in photoreceptors. We therefore propose that a specific amino acid transporter provides precursors for the synthesis of monoamine neurotransmitters. We further provide evidence that neurotransmitters can be synthesized de novo in a specific location.

## Results

### Hdc localizes to neuronal terminals

Histamine acts as major neurotransmitter at photoreceptor synaptic terminals, transmitting visual information to interneurons (*Hardie, 1989*). Further, histamine de novo synthesis in photoreceptor cells is essential for maintaining visual transmission (*Burg et al., 1993*). Interestingly, we have identified a vesicle transporter specific for histamine, LOVIT, which is concentrated exclusively in photoreceptor terminals and helps to maintain levels of histamine at synapses (*Xu and Wang, 2019*). Together with the fact that visual neurotransmission requires rapid and high-frequency firing, we hypothesize that the fast neurotransmitter histamine is synthesized directly in axon terminals. In *Drosophila* photoreceptor cells, histamine is initially synthesized from histidine by the eye-specific enzyme, Hdc. To determine the subcellular localization of Hdc in photoreceptor cells, we raised an antibody against an Hdc-specific peptide. Endogenous Hdc was detected predominantly in the lamina layer, which contains terminals of the R1–R6 photoreceptors, and in the medulla, which contains terminals of R7–R8. In these regions, Hdc co-localized with LOVIT and the presynaptic marker, CSP (cysteine string protein) (*Figure 1A–B*, and *Figure 1—figure supplement 1*). Moreover, Hdc was largely absent from

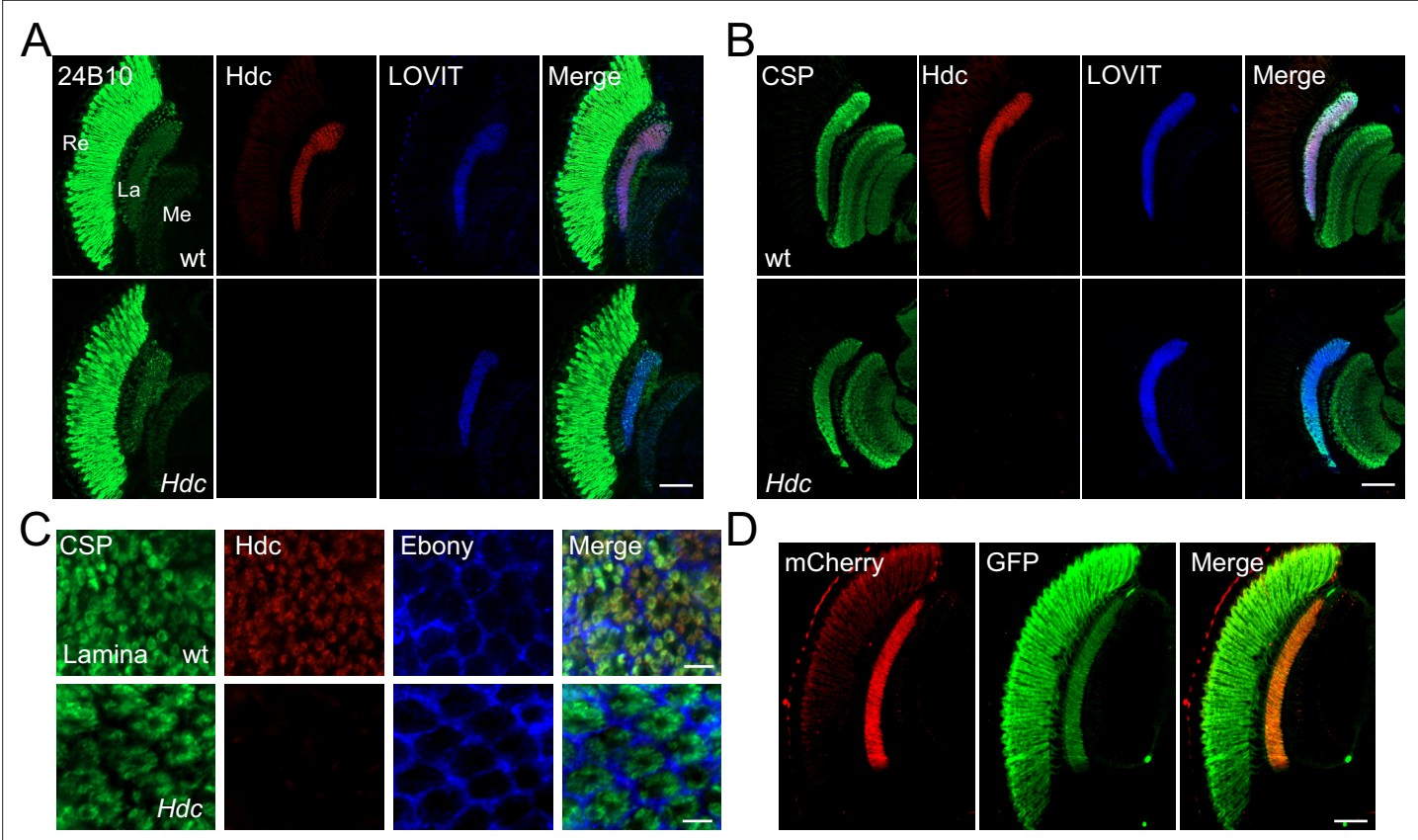

**Figure 1.** Histidine decarboxylase (Hdc) localizes to neuronal terminals. (**A–B**) Cryosections of *w1118* and *Hdc* mutant heads were labeled with antibodies against Hdc (red), 24B10 (**A**) (green, photoreceptor cell marker), CSP (cysteine string protein) (**B**) (green, localized to synaptic vesicles), and LOVIT (blue, labeling photoreceptor terminals). Scale bars, 50 µm. (**C**) Cross sections of the lamina layer showing overlapping patterns of Hdc (red) and CSP (green) localization, and a complementary pattern of Hdc (red) and Ebony (blue, expressed in lamina epithelial glia). Scale bars, 5 µm. (**D**) Cryosections of heads from *trp-Hdc-mCherry/trp-GFP* flies were labeled with antibodies against mCherry (red) and GFP (green). Scale bar, 50 µm. La, lamina; Me, medulla; Re, retina.

The online version of this article includes the following figure supplement(s) for figure 1:

**Figure supplement 1.** Histidine decarboxylase (Hdc) co-localizes with LOVIT in the medulla.

the retina (*Figure 1A and B*). Cross sections of the lamina neuropil that contained R1–R6 terminals revealed that Hdc was surrounded by the glial marker Ebony and co-localized with synaptic vesicle protein CSP (*Figure 1C*). To confirm the subcellular pattern of Hdc, we expressed mCherry-tagged Hdc in photoreceptor cells using the *trp* promoter (*transient receptor potential*) (*Montell and Rubin, 1989*). The chimeric Hdc was functional, as the *trp-Hdc-mCherry* transgene completely restored visual transmission in *Hdc* mutant flies. Consistent with what we observed for endogenous Hdc, Hdc-mCherry was also highly enriched in the lamina and medulla. In contrast, GFP signals were detected in the retina, lamina, and medulla in *trp-GFP* flies (*Figure 1D*). The finding that Hdc protein was enriched in photoreceptor terminals is consistent with the assumption that the neurotransmitter histamine is synthesized directly in axon terminals.

## TADR is required for visual synaptic transmission

Given that the enzyme responsible for catalyzing the biosynthesis of histamine localized to presynaptic regions, we next sought to determine how histidine, an Hdc substrate, is transported to neuronal terminals. We hypothesized that an amino acid transporter resided on the plasma membrane of photoreceptor synaptic terminals, and that this transporter would be responsible for histidine uptake and required for rapid histamine synthesis and visual transmission. Among ~600 putative transmembrane transporters encoded by the *Drosophila* genome (*Ren et al., 2007*), we identified 42 genes encoding SLC1, SLC6, SLC7, SLC17, SLC25, SLC32, SLC36, SLC38, and SLC66 families of proteins that were

predicted to import amino acids across the plasma membrane (*Bröer and Bröer, 2017*). These we tested as candidate histidine transporters (*Table 1*).

To examine whether these putative transporters were involved in visual neurotransmission, each candidate gene was knocked down individually via the eye-specific expression of RNAi using the *GMR (glass multiple response element)-Gal4* driver. Loss-of-function alleles were also used if available. We performed electroretinogram (ERG) recordings to determine which putative amino acid transporters functioned in visual transmission. ERG recordings are extracellular recordings that measure the summed responses of all retinal cells in response to light. An ERG recording from a wild-type fly contains a sustained corneal negative response resulting from photoreceptor depolarization, as well as ON and OFF transients originating from synaptic transmission to laminal LMCs (large monopolar cells) at the onset and cessation of light stimulation (*Wang and Montell, 2007*; *Figure 2A*). Flies deficient for histamine exhibited clear reductions in their ON and OFF transients, as shown for *Hdc^{P217}* mutant flies (*Burg et al., 1993*; *Melzig et al., 1996*). We found that knockdown of the putative cationic amino acid transporter (CAT) gene, *tadr*, resulted in the loss of synaptic transmission (*Ni et al., 2008*; *Figure 2A–B* and *Figure 2—figure supplement 1*). We then generated a different *tadr^{RNAi}* line, which we called *tadr^{RNAi-2}*. Consistent with the results of the screen, driving *tadr^{RNAi-2}* with *GMR-gal4* also affected ON and OFF transients (*Figure 2A and B*). Importantly, ERG transients were not affected by specific knockdown of *tadr* in glia using *repo-Gal4*, confirming the specific role of TADR in photoreceptor neurons (*Figure 2C and D*).

To further confirm that *tadr* was the causal gene, we generated a null mutation in the *tadr* gene by deleting a 665 bp genomic fragment using the CRISPR-associated single-guide RNA system (Cas9) (*Figure 2—figure supplement 2A*). PCR amplification and sequencing of the *tadr* locus from genomic DNA isolated from wild-type and *tadr²* flies revealed a truncated *tadr* locus in mutant animals, resulting in an out-of-frame fusion of exons 3 and 5 (*Figure 2—figure supplement 2B and C*). It has been reported that *tadr* mutation leads to photoreceptor degeneration. However, homozygous *tadr²* null mutants exhibited normal morphology of both the soma and axon terminal of photoreceptors when examined via transmission electron microscopy (TEM) (*Figure 2—figure supplement 3*). Furthermore, no retinal degeneration was observed in aged *tadr²* mutant flies. As a control, aged *culd¹* mutants exhibited extensive rhobdomere loss, which resulted from disruption of rhodopsin endocytic trafficking (*Xu and Wang, 2016*; *Figure 2—figure supplement 3A and B*). Consistent with the RNAi results, *tadr²* mutant flies displayed a complete loss of ON and OFF transients (*Figure 2E*). Further, expressing *tadr* in photoreceptors using the *trp* promoter restored ON and OFF transients in *tadr²* mutant flies, whereas expression of GFP failed to rescue the loss of ERG transients (*Figure 2E and F*). Disrupting visual transmission results in blindness, which reflected in the loss of phototactic behavior (*Behnia and Desplan, 2015*). We next used this behavioral assay to assess the vision of *tadr* mutant flies. Consistent with the ERG results, knockdown of *tadr* in the retina disrupted phototactic behavior, whereas wild-type levels of phototaxis were observed in flies in which *tadr* was knocked down in glia (*Figure 2G*). *tadr²* mutant flies also exhibited defective phototaxis, which was fully restored by the *trp-tadr* transgene (*Figure 2H*). Together, these findings reveal that TADR functions within photoreceptor cells to maintain synaptic transmission but not the integrity of neurons.

## TADR is a bona fide histidine transporter

Given that TADR belongs to a subfamily of CATs within the solute carrier 7 (SLC7) family (*Verrey et al., 2004*), we performed a histidine uptake assay in *Drosophila* S2 cells to determine whether TADR could transport histidine in vitro (*Han et al., 2017*; *Karl et al., 1989*). When Flag-tagged TADR was expressed in S2 cells the Flag-TADR signal localized exclusively to the plasma membrane (*Figure 3A*). We then transiently expressed TADR in S2 cells and assessed their ability to uptake [³H]-histidine. The histidine content of TADR-transfected cells was approximately 180 Bq/mg, which was 3.6-fold greater than measured for RFP-transfected controls (50 Bq/mg) (*Figure 3B*). Human SLC38A3, which is known to efficiently take up histidine, exhibited levels of histidine transport comparable to TADR, suggesting that TADR is a bona fide plasma membrane histidine transporter (*Bröer, 2014*; *Figure 3B*).

Considering that transporters related to histamine recycling are necessary for synaptic transmission, we next sought to determine whether TADR specifically transports histidine in *Drosophila*. Because a histamine transporter has not yet been identified, we first asked whether TADR can transport histamine. Histamine uptake assays revealed that TADR does not exhibit histamine uptake activity. As a

**Table 1.** Description of 42 putative amino acid transporters.

| CG number | On/OFF | Description |
|-----------|--------|-------------|
| CG4991 | Yes | Amino acid transmembrane transporter activity/SLC36A1 or A2 |
| CG7888 | Yes | Amino acid transmembrane transporter activity/SLC36A1 or A2 |
| CG1139 | Yes | Amino acid transmembrane transporter activity/SLC36A1 or A2 |
| CG8785 | Yes | Amino acid transmembrane transporter activity/SLC36A1 or A2 |
| CG32079 | Yes | Amino acid transmembrane transporter activity/SLC36A1 or A2 |
| CG32081 | Yes | Amino acid transmembrane transporter activity/SLC36A1 or A2 |
| CG16700 | Yes | Amino acid transmembrane transporter activity/SLC36A1 or A4 |
| CG13384 | Yes | Amino acid transmembrane transporter activity/SLC36A1 or A4 |
| CG43693 | Yes | Amino acid transmembrane transporter activity/SLC36A1 or A4 |
| polyph | Yes | Amino acid transmembrane transporter activity/SLC36A1 or A2 |
| path | Yes | Amino acid transmembrane transporter activity/SLC36A1 or A2 |
| mah | Yes | Amino acid transmembrane transporter activity/SLC36A1 or A2 |
| CG30394 | Yes | Amino acid transmembrane transporter activity/SLC38A10 |
| CG13743 | Yes | Amino acid transmembrane transporter activity/SLC38A11 |
| CG13248 | Yes | Amino acid transmembrane transporter activity/SLC7A4 |
| slif | Yes | Amino acid transmembrane transporter activity/SLC7A1 or A2 |
| CG12773 | Yes | Amino acid transmembrane transporter activity/SLC12A8 |
| NKCC | Yes | Amino acid transmembrane transporter activity/SLC12A3 |
| ChT | Yes | Amino acid transmembrane transporter activity/SLC5A7 |
| NAAT1 | Yes | Amino acid transmembrane transporter activity/SLC6A7 or A9 |
| CG15279 | Yes | L-amino acid transmembrane transporter activity/SLC6A7 or A9 |
| CG4476 | Yes | L-amino acid transmembrane transporter activity/SLC6A7 or A9 |
| CG1698 | Yes | L-amino acid transmembrane transporter activity/SLC6A7 or A9 |
| List | Yes | L-amino acid transmembrane transporter activity/SLC6A7 |
| JhI-21 | Yes | L-amino acid transmembrane transporter activity/SLC6A5 |
| mnd | Yes | L-amino acid transmembrane transporter activity/SLC7A6 or A7 |
| gb | Yes | L-amino acid transmembrane transporter activity/SLC7A6 or A7 |
| CG1607 | Yes | L-amino acid transmembrane transporter activity/SLC7A8 |
| sbm | Yes | L-amino acid transmembrane transporter activity/SLC7A9 |
| Eaat1 | Yes | L-amino acid transmembrane transporter activity/SLC7A3 |
| Eaat2 | Yes | L-amino acid transmembrane transporter activity/SLC7A2 |
| GC1 | Yes | L-glutamate transmembrane transport/SLC25A18 |
| tadr | No | Cationic amino acid transporter/SLC7A4 or SLC7A1 |
| VGlut | Yes | Vesicular glutamate transporter/SLC17A7 |
| Gat | Yes | GABA transporter activity/SLC6A1 |
| kcc | Yes | Potassium:chloride symporter activity/SLC12A4 |
| Sfxn1-3 | Yes | Serine transmembrane transporter activity/SFXN1 |
| VGAT | Yes | Vesicular GABA transporter activity/SLC32A1 |
| Ncc69 | Yes | Sodium:potassium:chloride symporter activity/SLC12A1 or A2 |

*Table 1 continued on next page*

*Table 1 continued*

| CG number | On/OFF | Description |
|---|---|---|
| CG1265 | Yes | Lysosomal amino acid transporter/SLC66A3 |
| CG3792 | Yes | Lysosomal amino acid transporter/SLC66 |
| CG13784 | Yes | Lysosomal amino acid transporter/SLC66A2 |

control, the human organic cation transporter (OCT2), which is known to take up histamine, exhibited high levels of histamine transport when expressed in S2 cells (**Busch et al., 1998**; **Figure 3C**). Next, we found that TADR did not exhibit β-alanine transporting activity when expressed in S2 cells, whereas BalaT, which served as a positive control, efficiently transported β-alanine (**Han et al., 2017**; **Figure 3D**). Similarly, TADR did not transport carcinine in S2 cells compared with the positive control, OCT2 (**Xu et al., 2015**; **Figure 3E**). To further determine if the amino acid transporter TADR is specific to histidine, we performed competition assays using [$^3$H]-histidine in combination with high concentrations of different L-amino acids or carcinine (0.5 mM for each L-amino acid or carcinine vs. 2.5 μM [$^3$H]-histidine). Histidine efficiently blocked [$^3$H]-histidine uptake, whereas the other amino acids and carcinine did not affect TADR-mediated histidine uptake. The only exception was lysine, which slightly reduced [$^3$H]-histidine uptake (**Figure 3—figure supplement 1**). These data support the conclusion that TADR is a specific histidine transporter involved in visual synaptic transmission.

## TADR predominantly localizes to photoreceptor terminals

Hdc and LOVIT mediate two steps critical for histamine synthesis and we found that both localize to axonal terminals. Thus, if TADR functions as a histidine transporter epistatic to Hdc, we should also detect TADR in photoreceptor axonal terminals. As we failed to generate a high affinity antibody against TADR, we used CRISPR/Cas9-based genome editing to introduce a GFP tag into the *tadr* locus (tagging the N-terminal), downstream of the native *tadr* promoter (*GFP-tadr*) (**Figure 4—figure supplement 1A** and Materials and methods). The reason for generating an N-terminal-tagged version of TADR is that expression of N-terminal-tagged but not C-terminal-tagged TADR fully restored synaptic transmission in *tadr^RNAi-2* flies. We identified *GFP-tadr* knock-in flies though PCR (**Figure 4—figure supplement 1B**). Importantly, homozygous *GFP-tadr* flies displayed intact ON and OFF transients, as expected, confirming that GFP-TADR retained in vivo function (**Figure 4—figure supplement 1C**). We found that TADR protein was enriched in photoreceptor cells including the retina, lamina, and medulla. Importantly, the GFP-TADR signal was concentrated in the lamina and medulla (marked with CSP), to which the R1–R6 and R7/R8 photoreceptors project their axons (**Figure 4A**). Moreover, both TADR and Hdc localized to terminals of photoreceptor neurons in the lamina and medulla (**Figure 4B**). These results demonstrated that TADR is expressed specifically in photoreceptor neurons and localized primarily to photoreceptor terminals. Further, the pattern of Hdc localization suggests that the de novo synthesis of a neurotransmitter occurs specifically at the relevant synapse.

## *tadr* mutants exhibit reduced levels of histamine at photoreceptor terminals

Given our evidence consistent with TADR-mediated transport of histidine directly into photoreceptor synaptic terminals, where histidine would be converted to histamine by Hdc, loss of TADR should reduce histamine levels, as has been seen for *hdc* mutants (**Borycz et al., 2000**). We generated head longitudinal sections from *tadr^2* mutants and wild-type controls and labeled them with an antibody reported to label histamine (**Chaturvedi et al., 2014**). In control flies, the histamine signal was enriched exclusively in photoreceptor terminals of both the lamina and medulla, co-localizing with CSP (**Figure 5A and C**). *tadr^2* mutants showed a dramatic reduction in histamine labeling at photoreceptor terminals in both the lamina and medulla (**Figure 5B and C**). Moreover, the apparent loss of histamine in *tadr^2* mutant photoreceptor terminals did not result from the loss of synaptic structures, as the density of synaptic vesicles and the number of capitate projections were comparable in the lamina of *tadr^2* mutant and control flies (**Figure 2—figure supplement 3**). We next used liquid chromatography-mass spectrometry (LC-MS) to examine in vivo levels of histamine in compound eyes of *tadr^2* mutant

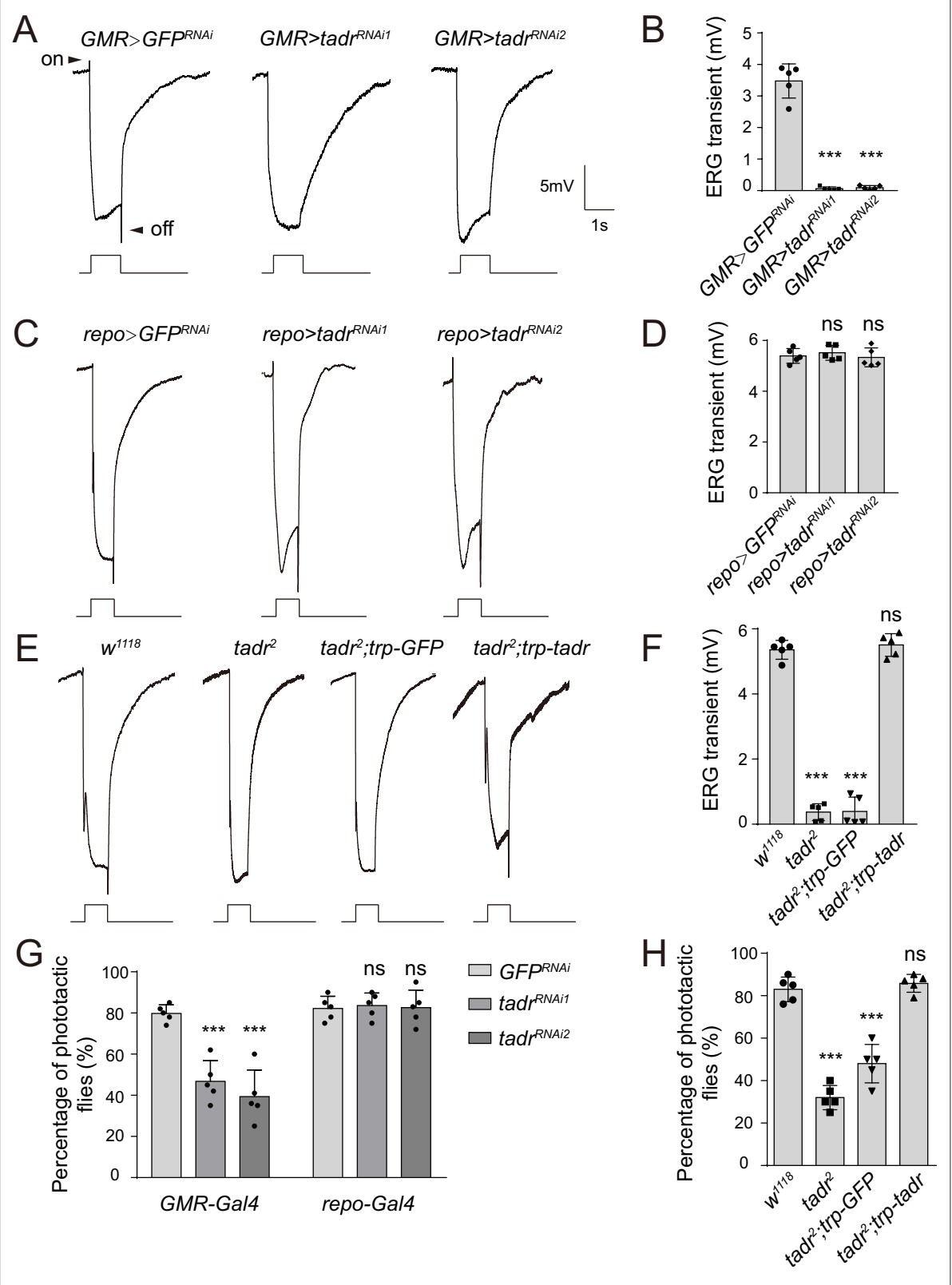

**Figure 2.** TADR (torn and diminished rhabdomeres) is required in photoreceptors for normal visual transmission. (**A–D**) Electroretinograms (ERGs) recorded from flies expressing various *UAS-tadr RNAi* transgenes (*tadr^RNAi1^* and *tadr^RNAi2^*) under the control of (**A**) *GMR-Gal4*, a driver specific for compound eyes (*GMR-Gal4/ UAS-tadr^RNAi1^* or *GMR-Gal4/+;UAS-tadr^RNAi2^/+*) and (**C**) the glial-specific driver *repo-Gal4* (*repo-Gal4/ UAS-tadr^RNAi1^* or *repo-Gal4/+;UAS-tadr^RNAi2^/+*). (**B**) Quantitative analysis of the amplitudes of ERG OFF transients shown in A compared with control flies (*GMR> GFP^RNAi^*, *GMR-*

*Figure 2 continued*

*Gal4/+;UAS-GFP^(RNAi)/+*) (one-way ANOVA; n = 5; ***p < 0.001). (**D**) Quantitative analysis of the amplitudes of ERG OFF transients shown in C compared with control flies (*repo> GFP^(RNAi)*, *repo-Gal4/UAS-GFP^(RNAi)*) (one-way ANOVA; n = 10; ns, not significant). Arrowheads indicate ON and OFF transients. One-day-old flies were dark adapted for 1 min and subsequently exposed to a 5 s pulse of orange light. (**E–F**) ERG recordings (**E**) and quantitative analysis of the amplitudes of ERG OFF transients (**F**) from wild-type (*w^1118*), *tadr^2*, *tadr^2;trp-GFP*, and *tadr^2;trp-tadr* flies. Displayed are comparisons to wild-type (*w^1118*) flies (one-way ANOVA; n = 10; ***p < 0.001; ns, not significant). (**G**) Phototactic behavior of flies corresponding to those in (**A**) and (**C**) compared with control flies (*GMR> GFP^(RNAi)* or *repo-> GFP^(RNAi)* flies). (**H**) Phototactic behavior of flies corresponding to those in (**E**). Each group is comprised of at least twenty 3-day-old flies. Five repeats were quantified for each group (one-way ANOVA, ***p < 0.001; ns, not significant).

The online version of this article includes the following source data and figure supplement(s) for figure 2:

**Source data 1.** Source data for quantitative of electroretinogram (ERG) transients and phototaxis behaviors.

**Figure supplement 1.** Electroretinogram (ERG) recordings for putative amino acid transporters functioned in visual transmission.

**Figure supplement 2.** Generation of *tadr^2* flies.

**Figure supplement 2—source data 1.** The full raw unedited gels for PCR products obtained from *c* mutants.

**Figure supplement 3.** Ommatidia and cartridges are normal in *tadr^2* mutants.

**Figure supplement 3—source data 1.** Source data for quantifying the average number of rhabdomeres per ommatidium at indicated days for wild-type (*w^1118*), *tadr^2*, and *culd^1* flies, as well as for quantifying capitate projection and synaptic vesicle density in *tadr^2* mutants.

flies. As expected, eyes from *Hdc^P217* mutant flies exhibited reduced levels of histamine, as these flies could not decarboxylate histidine into histamine (*Figure 5D*). Similarly, *tadr^2* mutants produced less histamine due to impairment of histidine uptake into photoreceptor terminals (*Figure 5D*). Moreover, histamine levels in *tadr^2* mutant flies were higher than in *Hdc^P217* mutant flies, suggesting that a small fraction of histidine could be supplied by other transporter systems. Consistent with previous reports, examining the heads of *Hdc^P217* mutant flies revealed less carcinine and β-alanine due to loss of histamine (*Melzig et al., 1998*; *Figure 5E and F*). Importantly, *tadr^2* mutants exhibited less carcinine and β-alanine as well, indicating that both TADR and Hdc are essential for the de novo synthesis of histamine. Moreover, reductions in histamine, carcinine, and β-alanine in *tadr^2* mutants were fully restored by expressing TADR in photoreceptor cells (*Figure 5E and F*). Reduction of histamine in *tadr^2* mutant flies indicates defective histamine synthesis and explains why photoreceptor synaptic transmission is disrupted in *tadr^2* mutants. These results therefore support the hypothesis that TADR transports histidine into photoreceptor terminals for the production of histamine to sustain tonic visual transmission.

## Ectopic expression of histidine transporters in photoreceptor cells rescues visual synaptic transmission in *tadr2* flies

If *tadr* mutants exhibit defective visual transmission because of deficient histidine transport, then replacing tadr with another transporter capable of transporting histidine should rescue photoreceptor synaptic transmission in *tadr* mutants. We first asked whether other members of the fly CAT family of transporters could efficiently transport histidine into S2 cell. We found that histidine was taken up by CG13248-transfected cells, but not by Slif (Slimfast)-positive cells (*Figure 6A*). Next, we overexpressed CG13248 or Slif in *tadr^2* mutant photoreceptor cells. The expression of CG13248 in *tadr^2* mutant photoreceptor cells fully restored both ERG transients and phototaxis, whereas Slif did not. These results are consistent with their abilities to transport histidine (*Figure 6B–D*). Further, expression of the human histidine transporter, SLC38A3, fully restored ERG transients and phototactic behavior in *tadr^2* mutant flies (*Figure 6A–D*). These data support an essential role for the histidine transporting activity of TADR in maintaining visual synaptic transmission. Taken together, we have identified a previously uncharacterized histidine transporter, TADR, and shown that TADR resides (together with the downstream enzyme, Hdc) in the axonal terminals of photoreceptor neurons, where it is responsible for the local biosynthesis of histamine.

## Discussion

Neurotransmitters are concentrated within presynaptic terminals and their release transmits signals to postsynaptic neurons. In most cases, the enzymes necessary for neurotransmitter synthesis are translated in the soma and then transported down the axon, where they then generate neurotransmitters. However, tyrosine hydroxylase and tryptophan hydroxylase, the rate-limited enzymes for dopamine

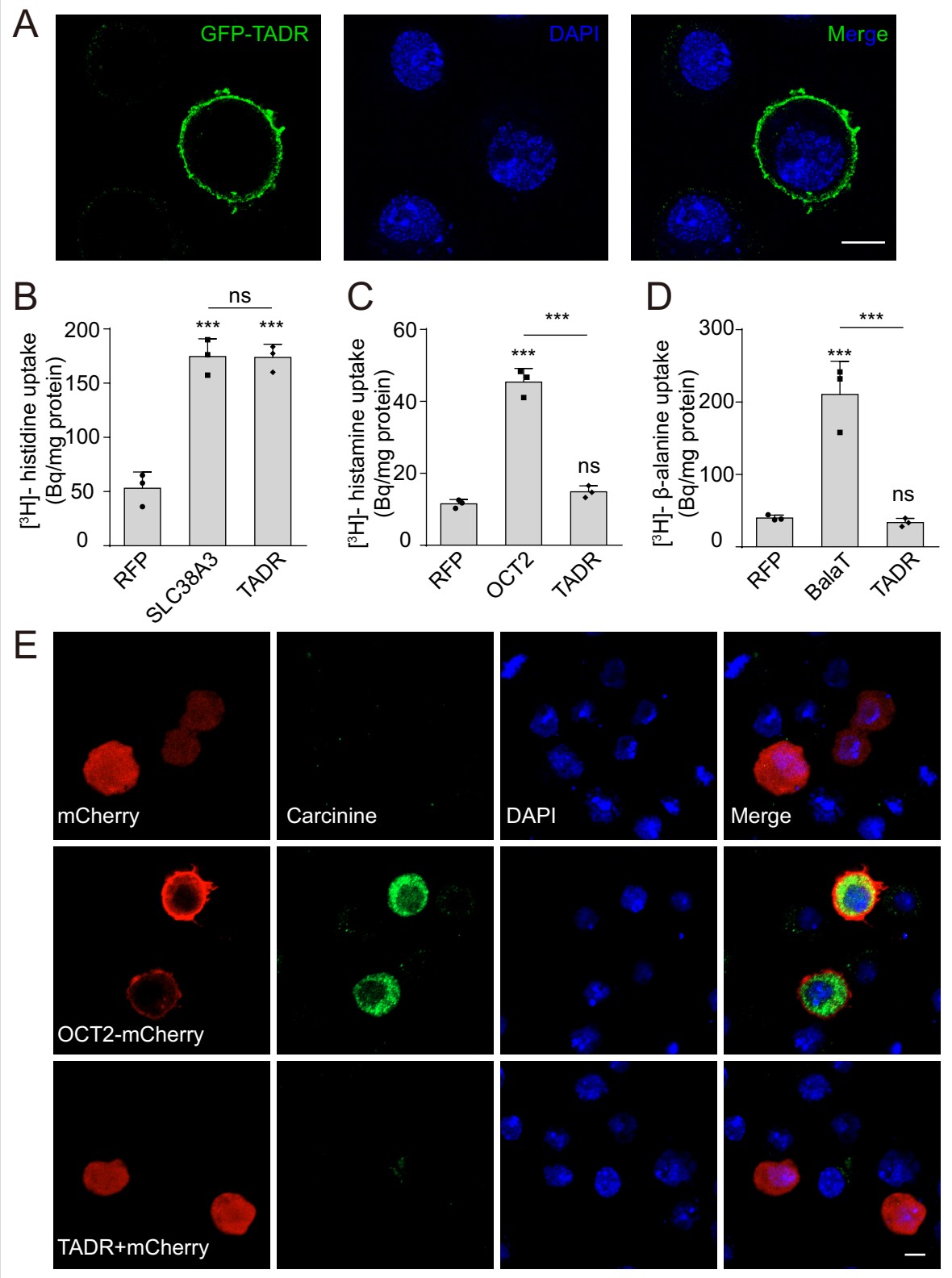

**Figure 3.** TADR (torn and diminished rhabdomeres) is a plasma membrane histidine transporter. (**A**) TADR localized to the plasma membrane when transiently transfected into S2 cells. GFP-tagged TADR was labeled with GFP antibody (green) and DAPI (blue), which stained the nucleus. Scale bar, 2 µm. (**B**) TADR transported histidine into S2 cells. Human SLC38A3 and RFP were used as positive and negative controls, respectively. [³H]-histidine was added to the DMEM solution (final concentration 2.5 µM). (**C–D**) TADR did not transport histamine (**C**) or β-alanine (**D**) into S2 cells. [³H]-histamine

*Figure 3 continued on next page*

*Figure 3 continued*

or [³H]-β-alanine was added to the ECF buffer (final concentration 3.7 × 10⁴ Bq), and organic cation transporter (OCT2) and BalaT served as positive transporter controls for histamine and β-alanine, respectively. Results are the mean ± SD of three experiments (one-way ANOVA, ***p < 0.001; ns, not significant). (**E**) TADR did not transport carcinine. S2 cells transiently expressing mCherry, OCT2-mCherry, or TADR/mCherry. Carcinine was added to the culture medium at a final concentration of 20 µM. Cells were labeled with rabbit anti-carcinine (green) antibody and DAPI (blue). The mCherry (red) signal was observed directly. Scale bar, 5 µm.

The online version of this article includes the following source data and figure supplement(s) for figure 3:

**Source data 1.** Source data for histidine, histamine, and β-alanine uptake assay.

**Figure supplement 1.** TADR (torn and diminished rhabdomeres) is a specific histidine transporter.

**Figure supplement 1—source data 1.** Source data for competition assays using histidine in combination with different L-amino acid or carcinine.

and serotonin synthesis, respectively, are cytosolic and reside both in the neuronal cell body and axon, suggesting that the de novo synthesis of these neurotransmitters occurs in both the soma and axon (*Cartier et al., 2010*; *Walther et al., 2003*). If neurotransmitters are generated in the cytosol of the cell body, the slow rate of diffusion for these small molecules could potentially limit the pool of axonal neurotransmitters and affect synaptic transmission. One possibility is that neurotransmitters are loaded into storage vesicles and that these vesicles are then taken to the nerve endings through fast axonal transport (*Broix et al., 2021*; *Roy, 2020*; *Vallee and Bloom, 1991*). However, the recently identified vesicular transporter, LOVIT, which is responsible for packaging histamine, is absolutely restricted to synaptic vesicles within the photoreceptor axon. Histamine is exclusively detected in wild-type photoreceptor terminals, but absent from *lovit* mutants (*Xu and Wang, 2019*).

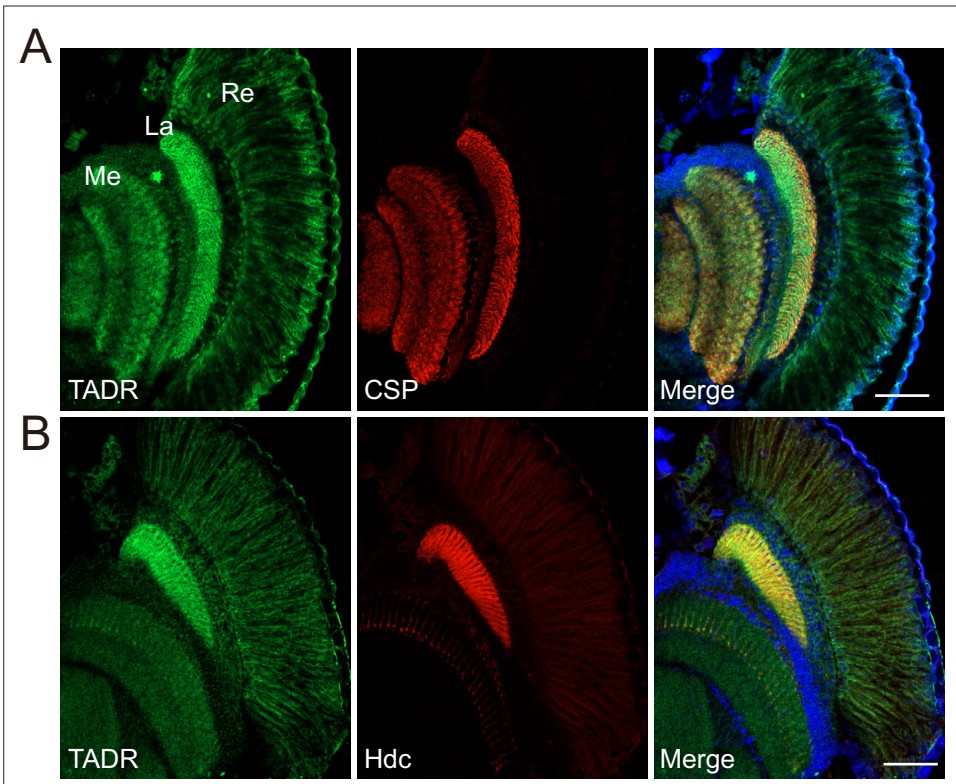

**Figure 4.** TADR (torn and diminished rhabdomeres) predominantly localizes to photoreceptor terminals. (**A–B**) Cryosections from *GFP-tadr* knock-in flies. Expression of an N-terminal GFP-tagged version of TADR was driven by the native *tadr* promoter. Sections were labeled for GFP-TADR with CSP (cysteine string protein) (red) (**A**) or Hdc (red) (**B**) and DAPI (blue). Scale bars, 50 µm. La, lamina; Me, medulla; Re, retina.

The online version of this article includes the following source data and figure supplement(s) for figure 4:

**Figure supplement 1.** Generation of *GFP-tadr* (torn and diminished rhabdomeres) flies.

**Figure supplement 1—source data 1.** The full raw unedited gels for PCR products obtained from *GFP-tadr* flies.

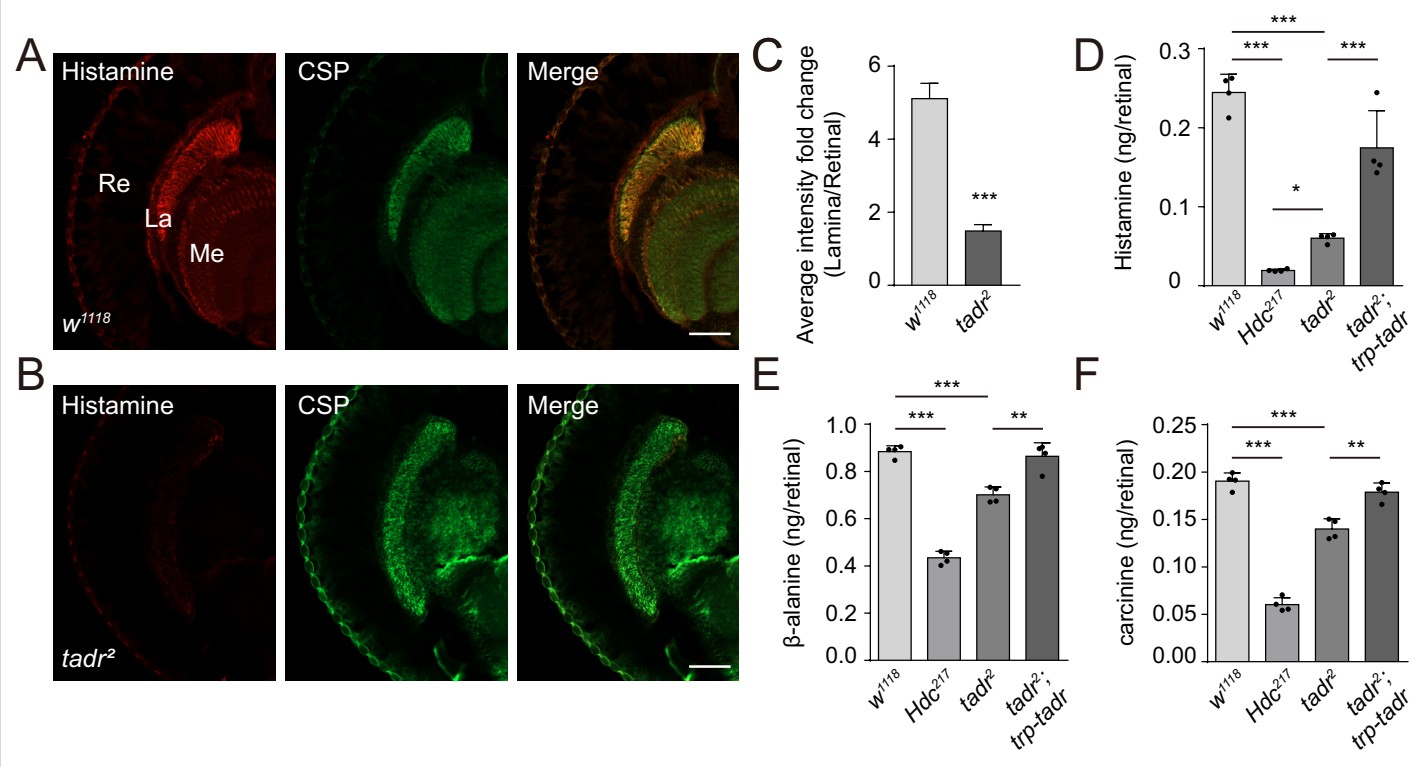

**Figure 5.** Loss of TADR (torn and diminished rhabdomeres) reduces histamine levels in vivo. (**A–C**) Histamine signaling in photoreceptor terminals was disrupted. Head cryosections were stained for histamine together with CSP (cysteine string protein) (synaptic vesicle marker) in control ($w^{1118}$) (**A**) and $tadr^2$ (**B**) flies. Sections are parallel to photoreceptor axons. Scale bars, 50 µm. (**C**) Average red fluorescence intensity ratio between the entire lamina and the retina immunolabeled layers. (**D–F**) Histamine (**D**), β-alanine (**E**), and carcinine (**F**) levels in compound eyes of 3-day-old control ($w^{1118}$), $Hdc^{P217}$, $tadr^2$, and $tadr^2;trp$-$tadr$ flies. Each sample included dissected compound eyes from 40 flies (one-way ANOVA; n = 4; ***p < 0.001; **p < 0.01; *p < 0.05). La, lamina; Me, medulla; Re, retina.

The online version of this article includes the following source data for figure 5:

**Source data 1.** Source data for the levels of histamine, β-alanine, and carcinine in $w^{1118}$, $Hdc^{P217}$, $tadr^2$, and $tadr^2;trp$-$tadr$ mutant fly compound eyes.

Consistent with this, we found that Hdc and its product histamine are enriched in photoreceptor terminals, suggesting that the de novo synthesis of histamine occurs exclusively in axons. If histidine, the substrate of Hdc, is transported to axonal terminals from the cell body, the rate of histamine biosynthesis would be limited. Therefore, it is possible that neurotransmitter precursors are taken up into terminals by specific transporters, and that neurotransmitter synthesis and packaging take place primarily within the axon. In support of this, we have characterized a new histidine transporter, TADR, which localized predominantly to photoreceptor terminals and was required for the de novo synthesis of histamine in photoreceptor terminals. Furthermore, TADR specifically transported histidine in vitro, and $tadr^2$ null mutant flies exhibited normal neuronal growth and survival, but disrupted visual transmission. Therefore, TADR and Hdc function synergistically within axonal terminals to provide a local pool of neurotransmitters (**Figure 6E**).

Although de novo synthesis provides a starting pool of neurotransmitters, recycling neurotransmitters after release is a critical pathway for maintaining neurotransmitter content within axon terminals. Ebony, the histamine recycling pathway involved in a previously identified *N*-β-alanyl-dopamine synthase, is expressed in epithelial glia and converts histamine to carcinine. The inactive histamine conjugate, carcinine, is transported back into photoreceptors where it is hydrolyzed back into histamine by Tan, an *N*-β-alanyl-dopamine hydrolase, to restore the neurotransmitter pool (**Borycz et al., 2002**; **Richardt et al., 2003**; **Richardt et al., 2002**; **Wagner et al., 2007**). Unlike Hdc, Tan localizes non-selectively to both the soma and axon, suggesting that the regeneration of histamine may take place in the soma as well (**Aust et al., 2010**). However, a recently identified transporter specific for carcinine, CarT, predominantly localizes to the terminals of photoreceptor neurons, rather than to the

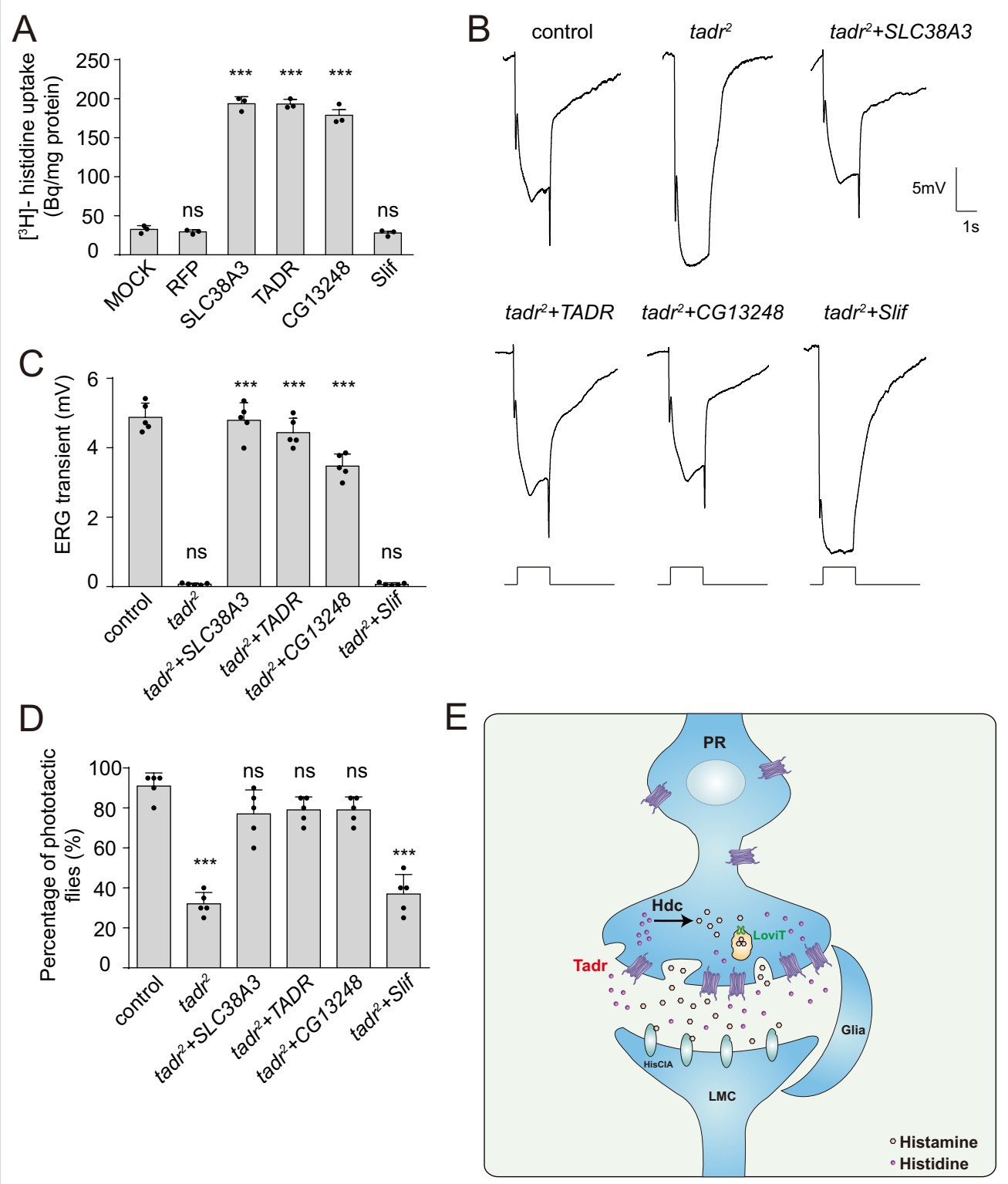

**Figure 6.** Rescue of defective visual transmission in *tadr²* mutants by expressing other histidine transporters. (**A**) SLC38A3 and CG13248 transported histidine into S2 cells, whereas the previously identified amino acid transporter, Slif, did not transport histidine. [³H]-histidine was added to the DMEM solution to a final concentration 2.5 µM. (**B**) Electroretinogram (ERG) recordings from control (*w¹¹¹⁸*), *tadr²*, *tadr²*+ *tadr* (*tadr²;longGMR-Gal4/UAS-tadr*), *tadr²*+ *CG13248* (*tadr²;longGMR-Gal4/UAS-CG13248*), *tadr²*+ *Slif* (*tadr²;longGMR-Gal4/UAS-Slif*), and *tadr²*+ *SLC38 A3* (*tadr²;longGMR-Gal4/UAS-SLC38A3*) are shown. Young flies (<3 days after eclosion) were dark adapted for 1 min and subsequently exposed to a 5 s pulse of orange light. (**C**) Quantitative analysis of the amplitude of ERG OFF transients shown in B. Displayed are comparisons to control (*w¹¹¹⁸*) flies (one-way ANOVA; n =

*Figure 6 continued on next page*

*Figure 6 continued*

10; \*\*\*p < 0.001; ns, not significant). (**D**) Phototactic behaviors of 3-day-old control, *tadr²*, *tadr²*+ *tadr*, *tadr²*+ *CG13248*, *tadr²*+ *Slif*, and *tadr²*+ *SLC38* A3 flies. Five repeats were made for each group, and each group had at least 20 flies (one-way ANOVA, \*\*\*p < 0.001; ns, not significant). (**E**) Model of the pathway for local histamine biosynthesis. Histidine is directly transported into photoreceptor cells at neuronal terminals by TADR, where it is used as a substrate to synthesize histamine by the decarboxylase, Hdc. Newly generated histamine is then loaded into synaptic vesicles by a LOVIT-dependent mechanism. Histamine, serving as a neurotransmitter, is released by photoreceptor cells (PR) to activate histamine-gated chloride channels (HisClA) on postsynaptic neurons (large monopolar cell [LMC]) to start visual transmission.

The online version of this article includes the following source data and figure supplement(s) for figure 6:

**Source data 1.** Source data for histidine uptake assay and source data for quantitative of electroretinogram (ERG) transients and phototaxis behaviors.

**Figure supplement 1.** Both CG13248 and Slif do not transport histamine.

**Figure supplement 1—source data 1.** Source data for the histamine uptake assay.

cell bodies, suggesting the regeneration of histamine from carcinine through axonal Tan (*Chaturvedi et al., 2016*; *Stenesen et al., 2015*; *Xu et al., 2015*). Taken together, both de novo synthesis and regeneration of the neurotransmitter histamine is restricted to neuronal terminals by a similar mechanism – the transportation of substrates.

TADR belongs to the CAT subfamily within the SLC7 family, and is homologous to the human membrane proteins, SLC7A4 and SLC7A1, which are predicted to be involved in importing basic amino acids across the plasma membrane (*Verrey et al., 2004*). It has been reported that flies carrying a missense mutation in *tadr*, namely *tadr¹* (generated from an EMS mutagenesis screen), exhibit retinal degeneration. This may indicate that TADR functions to provide amino acids to support photoreceptor cells (*Ni et al., 2008*). However, our *tadr²* null mutant flies fail to exhibit photoreceptor degeneration or growth defects. In addition, the morphologies of *tadr²* somas and axons are comparable to those of wild-type flies. In support of this, we found that TADR specifically transported histidine, with only a low affinity for lysine, suggesting that TADR is not involved in the general metabolism of amino acids. The neurodegeneration phenotype observed in *tadr¹* mutants may be due to other mutations generated by EMS mutagenesis or the neomorphic *tadr¹* point mutation, which may disrupt Gq signaling (*Ni et al., 2008*). The CG13248 transporter, a member of the CAT subfamily, is able to rescue the vision defects of *tadr²* mutant flies, but another member of the CAT subfamily, the transporter Slif, failed to do so. This is consistent with their ability to transport histidine (*Colombani et al., 2003*). Moreover, SLC38A3, which belongs to the SLC38 subfamily, is known to mediate sodium-dependent transport of multiple amino acids, including histidine, and the expression of SLC38A3 fully restores visual transduction and phototactic behavior in *tadr²* mutant flies (*Bröer, 2014*). The lack of sequence homology between SLC38A3 and TADR strongly suggests that TADR primarily acts as a histidine transporter to maintain visual transduction. Although TADR is a specific histidine transporter, its expression patterns suggests that it is not the only histidine transporter. Supporting this, null *tadr* mutants are viable and exhibit no growth or cell death phenotypes. If photoreceptor cells uptake histidine via other histidine transporters, they could have a small histidine pool and Hdc would be able to synthesize some histamine. In agreement with this, histamine levels in *tadr²* mutant are higher than in *Hdc* mutant, although histamine levels in both mutants are greatly reduced. Consistent with these relative histamine levels, *tadr²* mutation exhibit more phototactic behavior than *Hdc* mutants.

Amino acid transporters play fundamental roles in multiple metabolic processes, including mTOR activation, energy metabolism, nutritional stress, and tumor progression (*Chen et al., 2014*; *Colombani et al., 2003*; *Nicklin et al., 2009*; *Rebsamen et al., 2015*; *Wyant et al., 2017*). Consistent with these functions, the amino acid transporters SLC7A5 and SLC6A14 are upregulated in tumors, making then potential targets for the pharmacological treatment of cancer (*Kanai, 2022*; *Nałęcz, 2020*). Our experiments suggest that amino acid transporters provide amino acids that are critical for the de novo synthesis of monoamine neurotransmitters. Since inhibitors of monoamine transporters have been widely used as antidepressants, amino acid transporters specific for monoamine neurotransmitter synthesis (such as TADR) may provide new treatment options for neurological diseases associated with the dysregulation of monoamine neurotransmitters (*Andersen et al., 2009*).

# Materials and methods

**Key resources table**

| Reagent type (species) or resource | Designation | Source or reference | Identifiers | Additional information |
|---|---|---|---|---|
| Gene (*Drosophila melanogaste*) | *tadr* (cDNA) | *Drosophila* Genomics Resource Center | FLYB:FBcl0168145 | FlyBase symbol: LD25644 |
| Gene (*Drosophila melanogaste*) | *slif* (cDNA) | *Drosophila* Genomics Resource Center | FLYB:FBcl0167622 | FlyBase symbol: LD37241 |
| Gene (*Drosophila melanogaste*) | *CG13248* (cDNA) | *Drosophila* Genomics Resource Center | FLYB:FBcl0718746 | FlyBase symbol: FI04531 |
| Gene (*Homo sapiens*) | *SLC38A3* | NCBI database | Gene ID: 10,991 | Synthesized from GENEWIZ, China. |
| Genetic reagent (*Drosophila melanogaster*) | *tadr*[RNAi1] | Vienna *Drosophila* Resource Center | BDSC:v330472 RRID:FlyBase_ FBst0492187 | P{VSH330472}attP40 |
| Cell line (*Drosophila melanogaster*) | S2 | This paper | FLYB:FBtc0000181; RRID:CVCL_Z992 | Cell line maintained in N. Perrimon lab; FlyBase symbol: S2-DRSC. |
| Antibody | Anti-24B10(Mouse monoclonal) | DHSB | RRID:AB_528161 | IF(1:100) |
| Antibody | Anti-CSP(Mouse monoclonal) | DHSB | RRID:AB_528183 | IF(1:100) |
| Antibody | Anti-RFP(Ratmonoclonal) | Chromotek | RRID:AB_2336064 | IF(1:200) |
| Antibody | Anti-GFP(Rabbit polyclonal) | Invitrogen | RRID:AB_221569 | IF(1:200) |
| Antibody | Anti-LOVIT(Rat polyclonal) | *Xu and Wang, 2019* | | IF(1:100) |
| Antibody | Anti-Hdc(Rabbit polyclonal) | This paper | | IF(1:50) |
| Antibody | Anti-Ebony(Rabbit polyclonal) | University of Wisconsin | | IF(1:200) |
| Antibody | Anti-Histamine(Rabbit polyclonal) | ImmunoStar | RRID:AB_572245 | IF(1:100) |
| Recombinant DNA reagent | *pTrp-GFP* | This paper | | Used for generation of transgenic flies (maintained in T. Wang lab) |
| Recombinant DNA reagent | *pTrp-GFP-tadr* | This paper | | Used for generation of transgenic flies (maintained in T. Wang lab) |
| Recombinant DNA reagent | *pTrp-tadr-GFP* | This paper | | Used for generation of transgenic flies (maintained in T. Wang lab) |
| Recombinant DNA reagent | *pTrp-tadr* | This paper | | Used for generation of transgenic flies (maintained in T. Wang lab) |
| Sequence-based reagent | sgRNA1 (for *tadr*[2] mutant) | This paper | sgRNAs | GTGCCTGCGCTGCCCTGGCG |
| Sequence-based reagent | sgRNA2 (for *tadr*[2] mutant) | This paper | sgRNAs | TTTTAAGCGCCGTCGGCTGG |
| Sequence-based reagent | forward primer (for *tadr*[2] mutant) | This paper | PCR primers | CAATGGCAGGTGGGAGTTAGG |
| Sequence-based reagent | reverse primer (for *tadr*[2] mutant) | This paper | PCR primers | TTAGAGTCGCCGTGAATCGTC |
| Sequence-based reagent | sgRNA (for *GFP-tadr* knock-in) | This paper | sgRNAs | ACAACAACGACAATGTCGAG |
| Peptide, recombinant protein | Hdc peptide (*D. melanogaste*) | This paper | Synthesized by ChinaPeptides (Soochow, China) | CDFKEYRQRGKEMVDY |
| Chemical compound, drug | [$^3$H]-Histidine | American radiolabeled chemicals | ART 0234 | 30–60 Ci/mM |
| Chemical compound, drug | [$^3$H]-Histamine | American radiolabeled chemicals | ART 1432 | 10–40 Ci/mM |
| Chemical compound, drug | [$^3$H]-β-alanine | American radiolabeled chemicals | ART 0205 | 30–60 Ci/mM |
| Software, algorithm | GraphPad Prism software | GraphPad Prism (https://graphpad.com) | RRID:SCR_015807 | Version 7.0.0 |
| Software, algorithm | ImageJ software | ImageJ (http://imagej.nih.gov/ij/) | RRID:SCR_003070 | |

## Fly stocks and cultivation

The *Tl{Tl}Hdc^{attP}*, *Hdc^{P217}*, and *M(vas-int.Dm) ZH-2A;M(3xP3-RFP.attP) ZH-86Fb* flies were provided by the Bloomington *Drosophila* Stock Center (https://bdsc.indiana.edu). The *tdar^{RNAi1}* line (P{VSH330472} attP40) was obtained from the Vienna *Drosophila* Resource Center (https://stockcenter.vdrc.at). The transgenic RNAi lines for the in vivo transporter screen were obtained from the TsingHua Fly Center (http://fly.redbux.cn), the Bloomington *Drosophila* Stock Center, and the Vienna *Drosophila* Resource Center. The *w^{1118}*, *nos-Cas9*, *GMR-gal4*, and *Repo-Gal4* flies were maintained in the lab of Dr T. Wang at the National Institute of Biological Sciences, Beijing, China. Flies were maintained in 12-hr-light–12-hr-dark cycles with ~2000 lux illumination at 25°C, except when mentioned differently in the text.

## Generation of *tadr* mutant and knock-in flies

The *tadr^{2}* mutation was generated using the Cas9/sgRNA system (*Xu et al., 2015*). Briefly, two pairs of guide RNAs targeting the *tadr* locus were designed (sgRNA1: GTGCCTGCGCTGCCCTGGCG, sgRNA2: TTTTAAGCGCCGTCGGCTGG) and cloned into the *U6b-sgRNA-short* vector. The plasmids were injected into the embryos of *nos-Cas9* flies, and deletions were identified by PCR using the following primers: forward primer 5′-CAATGGCAGGTGGGAGTTAGG-3′ and reverse primer 5′-TTAG AGTCGCCGTGAATCGTC-3′. The *GFP-tadr* knock-in flies were generated as shown in *Figure 4—figure supplement 1*. Briefly, an sgDNA sequence (ACAACAACGACAATGTCGAG) was designed and cloned into the *U6b-sgRNA-short* vector. *tadr* genomic DNA (from 747 base pairs (bp) upstream of the transcription starting site to 893 bp downstream of the transcription termination site) was subcloned into a donor vector. GFP-tag sequence was then inserted at the end of the upstream fragment sequence. The two plasmids were co-injected into the embryos of *nos-Cas9* flies. The *GFP-tadr* flies were finally confirmed by PCR of genomic DNA using the following primers: forward primer 5′- ATGGTGAGCAAG-GGCGAGG –3′ and reverse primer 5′- GAATACCCACACACATGCCAATCA –3′. Both *tadr^{2}* and *GFP-tadr* flies were backcrossed to wild-type flies (*w^{1118}*) for two generations before preforming experiments.

## Generation of plasmid constructs and transgenic flies

Amino acid transporters including *tadr*, *slif*, and *CG13248* cDNA sequences were amplified from LD25644, LD37241, and FI04531 cDNA clones obtained from DGRC (*Drosophila* Genomics Resource Center, Bloomington, IN). The human SLC38A3 cDNA sequences were synthesized from GENEWIZ, China. Their entire CDS sequences were subcloned into the pCDNA3 vector (Invitrogen, Carlsbad, CA) for expression in HEK293T cells or PIB vector (Invitrogen, Carlsbad, CA) for expression in S2 cells. To expressing cDNAs in the photoreceptor cells, a 1.7 kb genomic DNA fragment (−1656 to +176 bp 5′to the transcription start site) of *trp* locus substituted the UAS sequence of *pUAST-attB* vector to generate the *pTrp-attB* vector (*Bischof et al., 2007*; *Li and Montell, 2000*). To construct *pTrp-tadr*, *pTrp-GFP-tadr,* and *pTrp-tadr-GFP*, the entire coding region of *tadr* was subcloned into the *pTrp-attB* vector with N-GFP or C-GFP-tags. To construct *pTrp-Hdc-mCherry*, the entire CDS sequence of *Hdc* with a C-terminal mCherry tag was cloned into the *pTrp-attB* vector. To construct *UAS-tadr*, *UAS-CG13248, UAS-Slif,* and *UAS-SLC38A3* plasmids, cDNAs of *tadr*, *CG13248*, *Slif,* and *SLC38A3* were amplified, and subcloned to *UAST-attB* vector. We produced a *tadr^{RNA2i}* line as described (*Ni et al., 2011*) by designing a 21-nucleotide short hairpin RNA sequences (GCCACAAGATGAGCAG CAAAT), and cloning them into a VALIUM20 vector. These constructs were injected into *M(vas-int. Dm) ZH-2A;M(3xP3-RFP.attP) ZH-86Fb* embryos, and transformants were identified on the basis of eye color. The *3xP3-RFP* and *w+* markers were removed by crossing with *P(Cre)* flies.

## Generation of anti-Hdc antibody

An Hdc peptide CDFKEYRQRGKEMVDY was synthesized by ChinaPeptides (Soochow, China), linked with BSA, and injected into rats by the Antibody Center at NIBS to generate anti-Hdc antibodies. The animal work for generating the antisera was conducted following the National Guidelines for Housing and Care of Laboratory Animals in China, and performed in accordance with institutional regulations after approval by the IACUC at NIBS (reference# NIBS2016R0001).

## ERG recordings

ERGs were recorded as described (*Xu et al., 2015*). Briefly, two glass microelectrodes were filled with Ringer's solution, and placed on the surfaces of the compound eye and thorax (one each surface). The

light intensity was ~0.3 mW/cm$^2$, and the wavelength was ~550 nm (source light was filtered using an FSR-OG550 filter). The electoral signals were amplified with a Warner electrometer IE-210, and were recorded with a MacLab/4 s A/D converter and Clampex 10.2 program (Warner Instruments, Hamden, CT). All recordings were carried out at 25°C.

## Histidine, β-alanine, histamine, and carcinine uptake assay

[$^3$H]-histidine (30–60 Ci/mM, American Radiolabeled Chemicals, St Louis, MO), β-alanine, [3-$^3$H (N)] (30–60 Ci/mM, American Radiolabeled Chemicals, St Louis, MO), and histamine [ring, methy-lenes-$^3$H(N)] dihydrochloride, (10–40 Ci/mM, American Radiolabeled Chemicals, St Louis, MO) uptake were measured as described (*Han et al., 2017*). Briefly, S2 cells were cultured in Schnei-der's *Drosophila* medium with 10% fetal bovine serum (Gibco, Carlsbad, CA) in 12-well plates, and transfected with vigofect reagent (Vigorous Biotechnology, Beijing, China). The transfected cells were washed with 1 mL extracellular fluid (ECF) buffer consisting of 120 mM NaCl, 25 mM NaHCO$_3$, 3 mM KCl,1.4 mM CaCl$_2$,1.2 mM MgSO$_4$, 0.4 mM K$_2$HPO$_4$, 10 mM D-glucose, and 10 mM Hepes (pH 7.4) at 37°C. Uptake assays were initiated by applying 200 μL DMEM (for histidine uptake) or ECF buffer (for histamine and β-alanine uptake) at 37°C. After 10 or 30 min, the reaction was terminated by removing the solution, and cells were washed with 1 mL ice-cold ECF buffer. The cells were then solubilized in 1 N NaOH and subsequently neutralized. An aliquot was taken to measure radioactivity and protein content using a liquid scintillation counter and a DC protein assay kit (Bio-Rad, Berkeley, CA), respec-tively. To perform histidine competition assays, [$^3$H]-histidine (30–60 Ci/mM, 2.5 μM) in combination with L-amino acid including serine, alanine, cysteine, glutamine, asparagine, arginine, and lysine, at higher concentration (0.5 mM) were added into DMEM buffer. Carcinine was added to the medium to yield a final concentration of 20 μM. After incubation for 3 hr, S2 cells were transferred to poly-L-lysinecoated slices, fixed with 4% paraformaldehyde, and incubated with rabbit anti-carcinine/hista-mine antibodies (1:100, ImmunoStar, Hudson, WI). Goat anti-rabbit lgG conjugated to Alexa 488 (1:500, Invitrogen, Carlsbad, CA) was used as secondary antibodies, and images were recorded with a Nikon A1-R confocal microscope.

## Immunohistochemistry

Fly head sections (10 μm thick) were prepared from adults that were frozen in OCT medium (Tissue-Tek, Torrance, CA). Immunolabeling was performed on cryosections sections with mouse anti-24B10 (1:100, DSHB), rat anti-LOVIT (1:100) (*Xu and Wang, 2019*), or anti-CSP (1:100, DSHB), rat anti-RFP (1:200, Chromotek, Germany), rabbit anti-Hdc (1:50), rabbit anti-GFP (1:200, Invitrogen, Carlsbad, CA), and rabbit anti-Ebony (1:200, lab of Dr S Carroll, University of Wisconsin, Madison, WI) as primary antibodies. For histamine immunolabeling, the rabbit anti-histamine (1:100, ImmunoStar, Hudson, WI) antibody was pre-adsorbed with carcinine, as previously reported (*Xu et al., 2015*). Goat anti-rabbit lgG conjugated to Alexa 488 (1:500, Invitrogen, Carlsbad, CA), goat anti-mouse lgG conjugated to Alexa 488 (1:500, Invitrogen, Carlsbad, CA), goat anti- rabbit lgG conjugated to Alexa 568 (1:500, Invi-trogen, Carlsbad, CA), and goat anti-rat lgG conjugated to Alexa 647 (1:500, Invitrogen, Carlsbad, CA) were used as secondary antibodies. The images were recorded with a Zeiss 800 confocal microscope.

## The phototaxis assay

Flies were dark adapted for 15 min before phototaxis assay, as described (*Xu et al., 2015*). A white light source (with an intensity of ~6000 lux) was used, and phototaxis index was calculated by dividing the total number of flies by the number of flies that walked above the mark. Five groups of flies were collected for each genotype, and three repeats were made for each group. Each group contained at least 20 flies. Results were expressed as the mean of the mean values for the four groups.

## Transmission electron microscopy

To visualize *Drosophila* retina ultrastructure, adult fly heads were dissected, fixed, dehydrated, and embedded in LR White resin (Electron Microscopy Sciences, Hatfield, PA) as described (*Xu et al., 2015*). Thin sections (80 nm) at a depth of 30–40 μm were prepared, and examined using a JEM-1400 transmission electron microscope (JEOL, Tokyo, Japan) equipped with a Gatan CCD (4k × 3.7k pixels, Palatine, IL). TEM of photoreceptor terminals was performed as described (*Xu and Wang, 2019*). Adult fly heads were dissected and fixed in 4% PFA. The laminas were further dissected by removing

retinas, followed by fine fixation in 1% osmium tetroxide for 1.5 hr at 4°C. Thin sections (80 nm) were stained with uranyl acetate and lead citrate (Ted Pella) and examined using a JEM-1400 transmission electron microscope (JEOL, Tokyo, Japan) equipped with a Gatan CCD (4k × 3.7k pixels, Palatine, IL).

### Liquid chromatography-mass spectrometry

LC-MS was performed as previous reported (*Han et al., 2017*). The Dionex Ultimate 3000 UPLC system was coupled to a TSQ Quantiva Ultra triple-quadrupole mass spectrometer (Thermo Fisher, Waltham, MA), equipped with a heated electrospray ionization probe in negative ion mode. Extracts were separated by a Fusion-RP C18 column (2 × 100 mm, 2.5 μm, phenomenex). Data acquired in selected reaction monitoring for histamine, carcinine, and β-alanine with transitions of 112/95.2, 183/95, and 90/72, respectively. Both precursor and fragment ions were collected with resolution of 0.7 FWHM. The source parameters are as follows: spray voltage: 3000 V; ion transfer tube temperature: 350°C; vaporizer temperature: 300°C; sheath gas flow rate: 40 Arb; auxiliary gas flow rate: 20 Arb; CID gas: 2.0 mTorr. Data analysis and quantification were performed using the software Xcalibur 3.0.63 (Thermo Fisher, CA). Each sample contained 50 *Drosophila* heads, and the mean values from five samples were calculated.

### Quantification and statistical analysis

All experiments were repeated as indicated in each figure legend. All statistical analyses were performed using GraphPad Prism 7. The variations of data were evaluated as mean ± SD. The statistical significance of the differences between two groups was measured by the unpaired two-tailed Student's t test, and one-way ANOVA or two-way ANOVA with Tukey's method, two-sided were performed for multi-group comparisons. A value of $p < 0.05$ was considered statistically significant (ns, not significant; $*p < 0.05$; $**p < 0.01$; $***p < 0.001$). p-Value, standard error of the mean (SD), and number are as indicated in each figure and legend.

## Acknowledgements

We thank the Bloomington Stock Center, *Drosophila* Genomic Resource Center, the Developmental Studies Hybridoma Bank, Vienna *Drosophila* Resource Center, TsingHua Fly Center and Dr S Carroll for stocks and reagents. We thank Y Wang and X Liu for assistance with fly injections. We are tremendously thankful for support provided by the Metabolomics Facility and Image Facility at NIBS. We thank State Key Laboratory of Membrane Biology, Institute of Zoology, Chinese Academy of Science for our Electron Microscopy and we would be grateful to Pengyan Xia for his help of taking EM images. We thank Dr D O'Keefe for comments on the manuscript. This work was supported by grants from the National Natural Science Foundation of China (81870693 and 81670891) awarded to T Wang.

## Additional information

### Funding

| Funder | Grant reference number | Author |
| --- | --- | --- |
| National Natural Science Foundation of China | 81870693 | Tao Wang |
| National Natural Science Foundation of China | 81670891 | Tao Wang |

The funders had no role in study design, data collection and interpretation, or the decision to submit the work for publication.

### Author contributions

Yongchao Han, Conceptualization, Data curation, Formal analysis, Investigation, Methodology, Validation, Visualization, Writing - original draft, Writing - review and editing; Lei Peng, Investigation, Methodology; Tao Wang, Conceptualization, Funding acquisition, Project administration, Supervision, Validation, Writing - review and editing

## Author ORCIDs

Yongchao Han ⓘ http://orcid.org/0000-0003-1965-0161
Tao Wang ⓘ http://orcid.org/0000-0002-2395-3483

## Ethics

The animal work for generating the antisera was conducted following the National Guidelines for Housing and Care of Laboratory Animals in China, and performed in accordance with institutional regulations after approval by the IACUC at NIBS (Reference# NIBS2016R0001).

## Decision letter and Author response

Decision letter https://doi.org/10.7554/eLife.75821.sa1
Author response https://doi.org/10.7554/eLife.75821.sa2

---

## Additional files

### Supplementary files

• Transparent reporting form

### Data availability

All data generated or analysed during this study are included in the manuscript and supporting file.

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
