## [Editor Report]

Han et al., report the discovery of an amino acid transporter that is required locally at axon terminals of fly photoreceptors neurons for the uptake of histidine, the precursor of the neurotransmitter histamine. This function is required for transmitter synthesis locally and neurotransmission. The work exemplifies a specialized model for local monoamine transmitter synthesis at synapses in the nervous system that may set the stage for tests of generality for other monoamine systems.

---

## [Decision Letter]

**Decision letter after peer review:**

Thank you for submitting your article "Tadr Is an Axonal Histidine Transporter Required for Visual Neurotransmission in *Drosophila*" for consideration by *eLife*. Your article has been reviewed by 3 peer reviewers, one of whom is a member of our Board of Reviewing Editors, and the evaluation has been overseen Claude Desplan as the Senior Editor. The following individual involved in review of your submission has agreed to reveal their identity: Helmut Krämer (Reviewer #2).

The reviewers have discussed their reviews with one another and all agreed on an invitation to resubmit based on the strengths of the manuscript. The Reviewing Editor has drafted this letter to help you prepare a revised submission.

Essential revisions:

1) Please provide a more detailed analysis of the difference between the previous allele that was shown to cause degeneration with the new one. Experimental additions would be welcome, but a new detailed analysis of the old allele should not be necessary.

2) Please see reviews 2 and 3 for some important experimental suggestions to substantiate conclusions, in particular:

– Despite the strong ERG phenotype, some 50% of the TADR mutant flies still show behavioral responses in the phototaxis axis, strongly arguing for a pathway acting in parallel to TADR. Comparison to a known blind mutant, such as HDC could clarify this issue.

– it is surprising that you did not test whether carcinine inhibits tadr and this would be useful to include in the manuscript.

*Reviewer #1 (Recommendations for the authors):*

I have only one specific question: The TADR-GFP knock-in shows widespread expression, yet the phenotypes seem to be very specific to the visual system, the only known system that uses histamine as a neurotransmitter in flies. The other components (hdc, lovit) are not expressed as widespread. Does this indicate another role for histidine uptake independent of histamine synthesis elsewhere in the nervous system?

*Reviewer #2 (Recommendations for the authors):*

Issues that need to be addressed:

1. The Figure legend for 1D indicates staining for Lovit, but it is not present in the image shown.

2. In the abstract and lines 107, 111, 211 the authors refer to HDC colocalizing with TADR, Lovit or CSP. None of the experiments referred to are up to modern standards to support co-localization. Instead, the authors should refer to these proteins both localizing to photoreceptor axons.

3. line 127: the authors should explain their criteria for identifying potential amino acid transporters.

4. line 138: The authors state: "Flies deficient for histamine exhibited clear reductions in their ON and OFF transients, as shown for Hdc[P217] mutant flies." This statement should be supported by data shown or reference(s) to papers that previously did show that.

*Reviewer #3 (Recommendations for the authors):*

The authors first use antibodies to Hdc and LOVIT to show enrichment in nerve terminals in both the lamina and medulla. They then perform an RNAi screen using the on/off transients of ERGS to detect changes in signaling from photoreceptors to laminar neurons and identify tadr. A supplemental yes/no table is provided for the other transporters. Given the online format of *eLife* it might be reasonable to include the raw data as a Supplementary file.

A brief explanation about how they chose specific transporters to test may also be useful since they tested 42 but state that there may be up to 600 candidates. It is helpful that they tested a second RNAi to confirm their results and used repo-Gal4 as a negative control. It is also impressive that they followed up the RNAi screen with a CRISPR mutant and further confirmed their results using both the mutant and rescue with a wt tadr transgene.

A brief statement about the previously published tadr1 mutant and whether it is or is not available would be useful.

They use in vitro assays in S2 cells to show uptake of histidine and show substrate specificity with a series of potential competitive substrates. It is surprising that they did not test carcinine (see below). Additional in vitro uptake assays show that tadr does not recognize β-alanine or histamine.

They tagged the tadr locus with GFP and demonstrate enrichment at nerve terminals as predicted. They show a reduction in histamine content in photoreceptor cell terminals in the mutant.

This was an excellent way to pull together in vitro and in vivo work.

Finally, they show rescue with another transporter that recognizes histamine will rescue the tadr mutant while a second transporter that does recognize histamine does not rescue ERGs. This is a clever experiment although it is not completely convincing without knowing what else CG13248 and Slif can transport, and whether there are other differences besides histidine.

The authors present a well-executed series of experiments and a convincing set of data supporting the idea that tadr is a histidine transporter essential for the function of photoreceptor cells. The manuscript nicely complements other papers from this group and others on transporters required for histaminergic signaling at photoreceptor cells. Indeed, this is an important paper for understanding synaptic signaling in the fly visual system and as such may be appropriate for *eLife*.

However, the first order synapse in the insect visual system is relatively unusual. To determine if tadr serves a more general role in histaminergic signaling, it would be useful to assess its localization in the CNS and PNS and possibly test other histamine dependent-behaviors such as grooming. It is possible that tadr is not expressed in other hdc(+) neurons in the fly and has nothing to do with grooming. This would not negate the importance of the paper. However, this information would seem to be an important component of characterizing tadr and whether it serves a more general role in histaminergic neurons. Labeling in the CNS would also help to establish whether tadr is specific for histidine. If so, it might not be expressed in non-histaminergic neurons.

It has been previously suggested that the histamine cycle involves conversion to carcinine via conjugation to β-alanine in glia prior to transport back to photoreceptors. The authors played an important role in this work. Tadr may have nothing to do with carcinine and the authors point out interesting differences in the localization of tan and tadr. Nonetheless, it is surprising that they did not test whether carcinine inhibited tadr and this would be useful to include in the manuscript. This is especially true since one of the key points of the paper is the unusual specificity of tadr As the authors note, most if not all amino acid transporters are less specific.

It is confusing that tadr1 but not tadr2 mutation caused rhabdomere degeneration. The authors should comment on this and provide possible reasons for this difference even if they are not able to obtain the tadr1 mutant. Is tadr2 simply a less severe allele because it has some residual activity? Were other genes disrupted in tadr1?

---

## [Author Response]

Essential revisions:1) Please provide a more detailed analysis of the difference between the previous allele that was shown to cause degeneration with the new one. Experimental additions would be welcome, but a new detailed analysis of the old allele should not be necessary.

We further examined retinal degeneration of *tadr^2^* mutants at different time point (1D, 5D, and 10D after eclosion). As shown in revised Figure 2—figure supplement 3, *tadr^2^* mutants did not exhibit any retinal degeneration phenotype, regardless of age. As a positive control, *culd^1^* flies exhibited obvious age-dependent retinal degeneration. We incorporated these data into the Results section (highlighted on Page 5).

2) Please see reviews 2 and 3 for some important experimental suggestions to substantiate conclusions, in particular:– Despite the strong ERG phenotype, some 50% of the TADR mutant flies still show behavioral responses in the phototaxis axis, strongly arguing for a pathway acting in parallel to TADR. Comparison to a known blind mutant, such as HDC could clarify this issue.

As an essential amino acid, each cell can only use extracellular histidine. Although TADR is a specific histidine transporter, its expression pattern suggests that it is not the only histidine transporter. Supporting this, null *tadr* mutants are viable and have no growth phenotype. As photoreceptor cells may uptake histidine from other histidine transporters, they have a small histidine pool and synthesize some histamine via Hdc. Supporting this, histamine levels in *tadr* mutants are higher than in *Hdc* mutants, although histamine levels are greatly reduced in both mutants (Figure 5D). Consistent with reduced histamine levels, *tadr^2^* mutants exhibit weak phototactic behavior, indicating the presence of another histidine transporter in *Drosophila* photoreceptor cells. However, given that *tadr^2^* mutants displayed a complete loss of ON and OFF transients, greatly reduced histamine levels, and much less phototactic behavior, we speculate that TADR is the major histidine transporter, responsible for maintaining the histidine pool and keeping visual transmission at high frequencies. We added this part to the Discussion section (Page 10 highlighted)

– it is surprising that you did not test whether carcinine inhibits tadr and this would be useful to include in the manuscript.

We thank the reviewer for this suggestion. In the previous version of the manuscript we showed that TADR failed to transport carcinine (Figure 3E of the previous version). In addition, we performed competition assays using [^3^H]-histidine in combination with carcinine at high concentration (0.5 mM for carcinine vs. 2.5 µM [^3^H]-histidine), and further confirmed that carcinine does not inhibit TADR activities. Please see the revised Figure 3—figure supplement 1B, and corresponding text on Page 6 for details (highlighted).

Reviewer #1 (Recommendations for the authors):I have only one specific question: The TADR-GFP knock-in shows widespread expression, yet the phenotypes seem to be very specific to the visual system, the only known system that uses histamine as a neurotransmitter in flies. The other components (hdc, lovit) are not expressed as widespread. Does this indicate another role for histidine uptake independent of histamine synthesis elsewhere in the nervous system?

We thank the reviewer for this concern. Hdc and LOVIT are exclusively involved in photoreceptor synaptic transmission and predominantly localize to neuronal terminals. By contrast, TADR is widely expression. As an essential amino acid, histidine is used for a variety of functions, including protein synthesis, proton buffering, metal ion chelation, scavenging of reactive oxygen, and erythropoiesis, in addition to the generation of histamine. Therefore, it is not surprising that the histidine transporter, TADR, is expressed more widely than Hdc and LOVIT. However, although TADR shows widespread expression, we found that TADR localizes predominantly to photoreceptor terminals, and the only phenotype we characterized is its functions in maintaining visual transmission at high frequencies.

Reviewer #2 (Recommendations for the authors):Issues that need to be addressed:1. The Figure legend for 1D indicates staining for Lovit, but it is not present in the image shown.

We apologize for this mistake and have deleted LOVIT from the figure legend. Moreover, we have made sure that all figure legends precisely match the relevant Figure.

2. In the abstract and lines 107, 111, 211 the authors refer to HDC colocalizing with TADR, Lovit or CSP. None of the experiments referred to are up to modern standards to support co-localization. Instead, the authors should refer to these proteins both localizing to photoreceptor axons.

We appreciate the reviewer for this suggestion and have changed the statement as suggested (highlighted on Pages 2 and 6)

3. line 127: the authors should explain their criteria for identifying potential amino acid transporters.

We chose the putative amino acid transporters based on the known solute carrier family of proteins in mammals that are involved in amino acid transport, including the SLC1, SLC6, SLC7, SLC17, SLC25, SLC32, SLC36, SLC38, and SLC66 families of proteins. We now explain in the manuscript that:

“we identified 42 genes encoding SLC1, SLC6, SLC7, SLC17, SLC25, SLC32, SLC36, SLC38, and SLC66 families of proteins that were predicted to import amino acids across the plasma membrane.”

We also added a review reference (Stefan Bröer and Angelika Bröer, PMID: 28546457) (highlighted on Page 4).

4. line 138: The authors state: "Flies deficient for histamine exhibited clear reductions in their ON and OFF transients, as shown for Hdc[P217] mutant flies." This statement should be supported by data shown or reference(s) to papers that previously did show that.

We added references after the statement as suggested. (highlighted on Page 5)

Reviewer #3 (Recommendations for the authors):The authors first use antibodies to Hdc and LOVIT to show enrichment in nerve terminals in both the lamina and medulla. They then perform an RNAi screen using the on/off transients of ERGS to detect changes in signaling from photoreceptors to laminar neurons and identify tadr. A supplemental yes/no table is provided for the other transporters. Given the online format of eLife it might be reasonable to include the raw data as a Supplementary file.

We thank the reviewer for this suggestion and have added these ERG results in the revised Figure 2—figure supplement 1.

A brief explanation about how they chose specific transporters to test may also be useful since they tested 42 but state that there may be up to 600 candidates. It is helpful that they tested a second RNAi to confirm their results and used repo-Gal4 as a negative control. It is also impressive that they followed up the RNAi screen with a CRISPR mutant and further confirmed their results using both the mutant and rescue with a wt tadr transgene.

We appreciate the reviewer for this suggestion. Previously, Ren et al., reported that ~600 putative transmembrane transporters were encoded by the *Drosophila* genome. We chose the putative amino acid transporters based on the known solute carrier family of proteins in mammals that are involved in amino acid transport, including the SLC1, SLC6, SLC7, SLC17, SLC25, SLC32, SLC36, SLC38, and SLC66 families of proteins. We now explain in the manuscript that:

“we identified 42 genes encoding SLC1, SLC6, SLC7, SLC17, SLC25, SLC32, SLC36, SLC38, and SLC66 families of proteins that were predicted to import amino acids across the plasma membrane.”

We also added a review reference (Stefan Bröer and Angelika Bröer, PMID: 28546457) (highlighted on Page 4).

A brief statement about the previously published tadr1 mutant and whether it is or is not available would be useful.

We thank the reviewer for this concern. A brief statement about the previously published *tadr^1^* mutant is now included in the manuscript. Please see our discussion for details (highlighted on Page 10).

They use in vitro assays in S2 cells to show uptake of histidine and show substrate specificity with a series of potential competitive substrates. It is surprising that they did not test carcinine (see below). Additional in vitro uptake assays show that tadr does not recognize β-alanine or histamine.

Thank you for this suggestion. In the previous version of the manuscript we show that TADR did not transport β-alanine and carcinine (Figure 3D and Figure 3E, respectively). In addition, we performed competition assays using [^3^H]-histidine in combination with carcinine at high concentration (0.5 mM for carcinine vs. 2.5 µM [^3^H]-histidine), and further confirmed that carcinine inhibits TADR activities. Please see the revised Figure 3—figure supplement 1B, and corresponding text on Pages 6 for details (highlighted). Because commercial radiolabeled carcinine is not available, we have tested carcinine uptake activity via immunostaining assays. These results shown that TADR did not transport carcinine in S2 cells compared with the positive control, OCT2. Please see Figure 3E, and corresponding text on Pages 6, 11, and 12 for details (highlighted).

They tagged the tadr locus with GFP and demonstrate enrichment at nerve terminals as predicted. They show a reduction in histamine content in photoreceptor cell terminals inthe mutant.This was an excellent way to pull together in vitro and in vivo work.Finally, they show rescue with another transporter that recognizes histamine will rescue the tadr mutant while a second transporter that does recognize histamine does not rescue ERGs. This is a clever experiment although it is not completely convincing without knowing what else CG13248 and Slif can transport, and whether there are other differences besides histidine.

TADR belongs to the cationic amino acid transporter (CAT) subfamily within the SLC7 family. We wondered whether other members of the fly CAT family of transporters could efficiently transport histidine into S2 cell. We found that histidine was taken up by CG13248, but not by Slif (Figure 6A). The expression of CG13248 in *tadr^2^* mutant photoreceptor cells fully restored both ERG transients and phototaxis, whereas Slif did not. To confirm the specific histidine uptake activity of CG13248, we performed additional histamine uptake assays. Neither CG13248 nor Slif showed histamine uptake activity. Please see the revised Figure 6—figure supplement 1, and corresponding text on Pages 13 and 14 for details (highlighted).

The authors present a well-executed series of experiments and a convincing set of data supporting the idea that tadr is a histidine transporter essential for the function of photoreceptor cells. The manuscript nicely complements other papers from this group and others on transporters required for histaminergic signaling at photoreceptor cells. Indeed, this is an important paper for understanding synaptic signaling in the fly visual system and as such may be appropriate for eLife.

We appreciate the reviewer for this statement.

However, the first order synapse in the insect visual system is relatively unusual. To determine if tadr serves a more general role in histaminergic signaling, it would be useful to assess its localization in the CNS and PNS and possibly test other histamine dependent-behaviors such as grooming. It is possible that tadr is not expressed in other hdc(+) neurons in the fly and has nothing to do with grooming. This would not negate the importance of the paper. However, this information would seem to be an important component of characterizing tadr and whether it serves a more general role in histaminergic neurons. Labeling in the CNS would also help to establish whether tadr is specific for histidine. If so, it might not be expressed in non-histaminergic neurons.

We thank the reviewer for this suggestion. It would be interesting to test other histamine dependent-behaviors such as grooming. Given that here we show that TADR-dependent local de novo synthesis of histamine is required for synaptic transmission in the *Drosophila* visual system, this additional analysis is out of the scope of this work. In the future, we plan to perform more widespread investigations to address more general roles for TADR in other histaminergic neurons.

It has been previously suggested that the histamine cycle involves conversion to carcinine via conjugation to β-alanine in glia prior to transport back to photoreceptors. The authors played an important role in this work. Tadr may have nothing to do with carcinine and the authors point out interesting differences in the localization of tan and tadr. Nonetheless, it is surprising that they did not test whether carcinine inhibited tadr and this would be useful to include in the manuscript. This is especially true since one of the key points of the paper is the unusual specificity of tadr As the authors note, most if not all amino acid transporters are less specific.

Thank you for this suggestion. We have performed competition assays using [^3^H]-histidine in combination with carcinine at high concentration (0.5 mM for carcinine vs. 2.5 µM [^3^H]-histidine). These data showed that carcinine did not affect TADR-mediated histidine uptake. Please see the revised Figure 3—figure supplement 1B, and corresponding text on Pages 6 and 13 for details (highlighted).

It is confusing that tadr1 but not tadr2 mutation caused rhabdomere degeneration. The authors should comment on this and provide possible reasons for this difference even if they are not able to obtain the tadr1 mutant. Is tadr2 simply a less severe allele because it has some residual activity? Were other genes disrupted in tadr1?

We thank the reviewer for this concern. We generated the *tadr^2^* mutation by deleting a 665-bp genomic fragment using the CRISPR-cas9 system, resulting in an out-of-frame fusion of exons 3 and 5 (deletion of 244 nt from the *tadr^2^* mRNA). Therefore, the *tadr^2^* mutant is a null allele. We added the molecular details of the mutant to the manuscript:

“PCR amplification and sequencing of the *tadr* locus from genomic DNA isolated from wild-type and *tadr^2^* flies revealed a truncated *tadr* locus in mutant animals, resulting in an out-of-frame fusion of exons 3 and 5 (Figure 2—figure supplement 2B and 2C).” (Page 5, highlighted).

We also further examined retinal degeneration of *tadr^2^* mutants at different time points (1, 5, and 10 days after eclosion). As shown in the revised Figure 2—figure supplement 2, *tadr^2^* mutants did not exhibit any retinal degeneration, regardless of age. As a positive control, *culd^1^* flies exhibited obvious age-dependent retinal degeneration. We incorporated these data into the Results section (highlighted on Page 5). Since the *tadr^2^* mutation is a null *tadr* allele, TADR is not required to maintain photoreceptor integrity. In contrast to *tadr^2^*, *tadr^1^* is a point mutation generated via EMS mutagenesis. As EMS creates random mutations throughout the genome, mutations in other loci may cause the retinal degeneration phenotype. It is also be possible the *tadr^1^* mutant is a neomorphic allele. We explain this point in the manuscript as:

“The neurodegeneration phenotype observed in *tadr^1^* mutants may be due to other mutations generated by EMS mutagenesis or the neomorphic *tadr^1^* point mutation, which may disrupt Gq signaling.”

Please see our discussion for details. (highlighted on Page 10).